

# Creep enhancement and sliding in a temperate, hard-bedded alpine glacier

Juan P. Roldán-Blasco[1], Adrien Gilbert[1], Luc Piard[1], Florent Gimbert[1], Christian Vincent[1], Olivier Gagliardini[1], Anuar Togaibekov[1,2], Andrea Walpersdorf[2], and Nathan Maier[1]

[1]IGE, Univ. Grenoble Alpes, CNRS, INRAE, IRD, Grenoble INP, 38000 Grenoble, France
[2]ISTerre, Univ. Grenoble Alpes, CNRS, IRD, UGE, 38000 Grenoble, France

**Correspondence:** Adrien Gilbert (adrien.gilbert@univ-grenoble-alpes.fr)

**Abstract.** Glacier internal deformation is usually described by Glen's Law using two material parameters, the creep factor $A$ and the flow law exponent $n$. However, the values of these parameters and their spatial and temporal variability are rather uncertain due to the difficulty of quantifying internal strain and stress fields at the natural scale. In this study, we combine 1-year long continuous measurements of borehole inclinometry and surface velocity with three-dimensional full Stokes ice

flow modeling to infer ice rheology and sliding velocity in the ablation zone of the Argentière Glacier, a temperate glacier in the French Alps. We demonstrate that the observed deformation rate profile has limited sensitivity to the flow law exponent $n$ and instead mainly reflects an increase in the creep factor $A$ with depth, with $A$ departing from its surface value by at least up to a factor of 2.5 below 160 m. We interpret this creep factor enhancement as an effect of increasing interstitial water content with depth from 0% to 1.3% which results in an average value of $A = 148$ MPa$^{-3}$ a$^{-1}$. We further observe that internal ice

deformation exhibits seasonal variability similar to that in surface velocity, such that the local basal sliding velocity exhibits no significant seasonal variation. We suggest that these changes in deformation rate are due to variations in the stress field driven by contrasting changes in subglacial hydrology conditions between the side and the center of the glacier. Our study gives further evidence that borehole inclinometry combined with full-Stokes flow model allows constraining both ice rheology and basal friction at scales that cannot be inferred from surface velocity measurements alone.

## 1 Introduction

Glacier dynamics depends on both internal deformation and basal sliding. Given the scarcity of direct observations of ice rheology and basal sliding speed at the natural scale, ice flow models commonly use inverse methods to estimate the material parameters that allow the best fit to the surface velocity (e.g. Arthern and Gudmundsson, 2010; Fürst et al., 2015; Mosbeux et al., 2016). However, the problem is largely undetermined due to poor knowledge of ice thickness and does not allow the

identification of model weaknesses, as model errors are compensated by material parameter adjustments. Independent and accurate estimates of ice material parameters as well as sub-glacial basal friction changes are key in order to better represent glacier dynamics in models, in particular the fraction of surface velocities that is due to basal sliding versus internal ice deformation.



Ice deformation is known to follow Glen's flow law (Glen, 1955), with the creep parameter $A$ being primarily dependent on
ice temperature (Barnes et al., 1971; Weertman, 1983). Such dependency has been studied extensively in both the laboratory and
the field (Cuffey and Paterson, 2010), and can reasonably well be accounted for in models given a temperature profile. Typical
values for ice viscosity and exponents for temperate glaciers and ice-caps are $A \approx 75$ MPa$^{-3}$ a$^{-1}$ and $n = 3$, respectively
(Cuffey and Paterson, 2010). The applicability of these values to describe glacier dynamics at the natural scales, however, is still
quite uncertain, mainly as a result of other controls coming into play, such as ice texture (orientation and microstructure, (Barnes
et al., 1971; Goldsby and Kohlstedt, 2001)), impurities (Jones and Glen, 1969) or water content (Lliboutry and Duval, 1985;
Duval, 1977). These controls are challenging to evaluate as they may vary quite extensively in time and space depending on
stress and deformation conditions (Chauve et al., 2024; Rathmann and Lilien, 2022). This is especially true under temperate ice
conditions, which are hard to investigate in the laboratory where water content is difficult to control, but also hard to characterize
in the field, since radar observations traditionally used to recover ice texture (Young et al., 2021) and potentially water content
(Ogier et al., 2023) are difficult to interpret due to large wave scattering under temperate ice conditions. Nevertheless, in many
settings, temperate ice deformation is expected to primarily control ice dynamics, including on ice-sheets like Greenland where
most of the deformation concentrates in the basal temperate layer (Law et al., 2023).

Basal sliding on hard beds is also known to be a function of ice deformation as it is enhanced near the bed, thus with the
same sources of uncertainty as presented above, but with the added complexity that subglacial hydrology also affects it. Water
pockets, commonly referred to as cavities, can form in the lee side of bedrock bumps, reducing the apparent bed roughness,
facilitating creep and thus enhancing basal sliding (Lliboutry, 1959, 1968). Subglacial channels can also form under sufficient
turbulent-induced melt, causing the opposite effect of reducing the basal sliding through lowering the basal water pressure
(Röthlisberger, 1972; Schoof, 2010). Ultimately, the resultant effect of subglacial hydrology on overall glacier dynamics thus
depends on the type of subglacial hydrology system at play at any given time and location. Of fundamental importance is the
ability to evaluate the spatio-temporal evolution of basal sliding to improve our representation of the evolution of subglacial
hydrology and its link to overall glacier dynamics in glacier and ice-sheet models.

A unique means to investigate ice deformation and basal sliding simultaneously is through borehole inclinometry, either by
measuring the change in borehole orientation through repeated surveys (e.g. Perutz, 1949; Shreve and Sharp, 1970; Raymond,
1971; Hooke, 1973; Hooke and Hanson, 1986; Hooke et al., 1992; Harper et al., 2001; Marshall et al., 2002; Chandler et al.,
2008) or through continuous englacial tiltmeter recording (e.g. Gudmundsson et al., 1999; Lüthi et al., 2002; Willis et al.,
2003; Amundson et al., 2006; Ryser et al., 2014; Keller and Blatter, 2012; Doyle et al., 2018; Lee et al., 2019; Maier et al.,
2019, 2021). These observations provide deformation rate profiles that can allow in-situ estimates of the ice rheology, basal
velocity and their respective temporal variations, with basal velocity obtained through integrating the deformation rate profile
with depth and removing it from the surface velocity obtained from GPS (Hooke et al., 1992; Maier et al., 2021). However,
a major challenge in evaluating ice rheology based on these observations is the retrieval of the stress field against which de-
formation rates can be compared. Since stresses cannot be measured, they must be estimated independently using a modeling
approach, which, depending on model assumptions or prescribed boundary conditions, can introduce large uncertainties in
the derived creep factor or flow exponent, especially in valley glaciers (e.g. Harper et al., 2001; Chandler et al., 2008) where



the highly three-dimensional geometry requires an evaluation of the full stress tensor. Even on large ice caps, strong spatial

variations in the measured deformation rate between different boreholes have been shown to reflect a complex stress field influenced by spatial variability in bed friction (Ryser et al., 2014), which, together with variations in ice temperature, complicates accurate assessment of ice rheology. The stress field complexity can also influence the temporal variation that can be observed in deformation rate either due to flow over a changing bed topography (Maier et al., 2019) or due to hydrologically-driven temporal changes in basal drag patterns (Hooke et al., 1992; Willis et al., 2003).

In this paper, we attempt to infer internal rheological parameters and reconstruct basal velocity through time by combining continuous borehole inclinometry observations with full Stokes 3-dimensional modeling of the stress field over a full melting season. The study focuses on the Argentière Glacier (temperate ice, French Alps), which has been intensively monitored for several decades (mass balance, surface velocities and topography, sliding velocity, bedrock geometry) and provides a unique and well-constrained environment for a natural scale study of ice deformation. We first describe the study site and the mea-

surement methods. We then analyse the observation in terms of material parameters using the flow model and finally provide the observed time series of both deformation and basal velocities. With this methodology we identify a depth-dependency of ice viscosity, which we attribute to changes in interstitial water content, as well as temporal changes in ice deformation, which we argue are due to subglacial-hydrology driven changes in basal friction conditions. These novel observations contribute to a better understanding of the complex interplay between basal sliding and internal ice deformation, while providing a rare

constraint on ice rheology in a natural setting.

## 2    Field site and instrumentation

### 2.1    The Argentière Glacier

The Argentière Glacier is a temperate glacier located in the Mont Blanc range, French Alps (45°10 N, 6°10 E). The glacier rests on a hard bedrock (Vivian and Bocquet, 1973) and extends for 9 km within an altitude range of 1600 m to 3400 m, separated

by an icefall at 2300 m. The dynamics of the glacier has been continuously monitored since the 1970's, in particular its basal sliding velocities, thanks to direct access to the glacier bed  700 m downstream of the drilling site (Vincent and Moreau, 2016; Gimbert et al., 2021a; Gilbert et al., 2022). The measurements are made by a cavitometer installed in a natural ice cavity that records sliding velocity at a half-hour resolution. Surface dynamics shows a seasonal pattern typical of mountain glaciers, with low velocity between September and April, followed by a period of sustained high velocity between May and August (Vincent

et al., 2022). The glacier also benefits of subglacial runoff monitoring by the power company Electricité d'Emosson SA and continuous records of yearly surface mass balance, topography and velocity from the GLACIOCLIM monitoring program (https://glacioclim.osug.fr/). Several campaigns of ground penetrating radar measurement also provide a good knowledge of the basal topography (Rabatel et al., 2018; Sergeant et al., 2020; Gimbert et al., 2021b).





**Figure 1.** (a) Map of the ablation area of the Argentière Glacier (projection EPSG:27572) with ice thickness (black contours) and instrument locations. (b) Estimated initial shape, drilled depth (black triangles) and instrumented depth (every star is a tiltmeter) of the five boreholes. (c) to (f) Tilt $\theta$ (continuous lines) recorded at four example inclinometers in BH2 (numbers are shown on panel (b)). The dashed vertical lines mark the 15th of February 2020, the day when we start our analysis.

## 2.2 Borehole deformation instrumentation

The drilling sites are located in the central part of the ablation area, between 600 and 800 m upstream of the icefall at an elevation of 2380 m. The thickness at the center flow line of this area is about 230 - 250 m and the bedrock forms an over-





| Borehole | Tiltmeters (nb) | Borehole depth (m) | Instrumented depth (m) | Bedrock depth (m) |
|----------|-----------------|--------------------|------------------------|-------------------|
| BH1 | 18 | 208 | 190 | 253±10 |
| BH2 | 19 | 238 | 234 | 237±20 |
| BH3 | 17 | 216 | 174 | 235±20 |
| BH4 | 19 | 237 | 211 | 234±20 |
| BH5 | 17 | 194 | 190 | 234±10 |

**Table 1.** Summary of the boreholes instrumentation and depths after installation. The instrumented depth refers to the depth of the last sensor. The bedrock depth is the one previously estimated by ground penetrating radar measurements (Rabatel et al., 2018; Sergeant et al., 2020)

deepening where most of the boreholes are located, see Figure 1a and Table 1. The average surface speed at this location is about 47 m a$^{-1}$ (Vincent et al., 2022).

Drilling operations took place between the 12th and the 14th September, 2019. We used a custom-built hot water drill
operating at 70°C to drill 10 cm diameter boreholes at an average speed of 60 m/h. Insufficient weight of the driller head, fast drilling speeds, and intraglacial debris affected the verticality of the boreholes. In several occasions, probably due to the presence of rocks inside the glacier (Hantz and Lliboutry, 1983), the driller head would stop advancing, enlarging the size of the borehole at the location until drilling could be resumed. On a few occasions we could not pursue drilling and a new location had to be chosen. There were instances of sudden borehole drainage, indicating that the borehole was connected with
a water pathway or a crevasse. The position of the final completed and instrumented boreholes (BH1, BH2, BH3, BH4, BH5) are given by the blue dots in Figure 1a. We estimate the initial shape of the boreholes calculated from the tilt and azimuth data approximately one month after installation to correct for borehole and sensor depth errors due to boreholes not being perfectly vertical (Figure 1b). Note that the actual shapes are 3D curves, so we instead show the estimated horizontal distance between each inclinometer and a vertical line starting at the surface.
The deformation rate sensors consist of a custom printed circuit board (PCB) equipped with a high-end triaxial gravity sensor (Muratta SCL3000) and a triaxial magnetic sensor (STMicroelectronics LSM303). The sensors are each connected to a microcontroller (Atmega 328P) via a Serial Peripheral Interface (SPI) and Inter-Integrated Circuit (I2C) bus, and then soldered to the PCB. PCB manufacturing and component soldering were outsourced, before each microcontroller was programmed and calibrated in the laboratory. To withstand the pressure of moving ice and water pressure, the PCB was encapsulated with an
epoxy compound inside an aluminum tube with an outer diameter of 25 mm, which we call the tiltmeter. The sensors are connected to a surface unit consisting of a Campbell Scientific data logger (CR300), two 12V 55Ah gel batteries and a solar panel, which allows autonomous data acquisition. Communication between the inclinometers and the surface unit is via the Modbus communication protocol over half-duplex RS485 serial buses. The gravity sensors are used to determine the position of the sensor with respect to its own reference system, from which we can derive the tilt $\theta$, the angle with respect to the vertical,
with an estimated accuracy of 0.01°. The lab calibration has shown that tilt readings above 45° become increasingly unreliable. The magnetic sensors do not provide good absolute measures of orientation relative to North because they are very sensitive to





parasitic magnetic fields. For this reason, we decide not to use them except to roughly estimate the initial shapes of the holes (Figure 1b).

The tiltmeters are grouped in chains of 20, more densely concentrated towards the bottom of the glacier, see Figure 1b. For
each borehole $i$, we name each tiltmeter $j$ as BH$i$#$j$, starting with 1 for the deepest tiltmeter, i.e. BH2#5 is the fifth deepest tiltmeter installed in the second borehole. All sensors acquired data every 30 minutes.

Tiltmeter array performance varied between boreholes. The sensor array in BH1 stopped working after a few days and provided no useful data. In BH2, all sensors recorded data for more than a year until late October 2020, when the cable snapped at an estimated depth of 220 meters, losing the deepest 6 meters. BH3 and BH4 were drilled close to BH2, with BH3 being
the shortest hole of the entire campaign. Both BH3 and BH4 show a very crooked shape in their deepest tiltmeters, suggesting problems during drilling that affect the quality of the measurements, either by having non-vertical tiltmeters that are then too sensitive to normal strain (Keller and Blatter, 2012), or by having poor sensor-ice mechanical coupling as a result of too wide borehole diameter. All sensors in BH3 and BH4 worked until November 2020. The sensors in BH5 stopped working soon after installation. The number of tiltmeters installed in each borehole depends on the borehole depth. We provide a summary of
borehole and sensor corrected depths in Table 1. The bedrock depth at BH2, BH3 and BH4 is based on Sergeant et al. (2020), which suggests that the DEM derived ice thickness in the vicinity of BH4 (Rabatel et al., 2018) is underestimated by about 20 m. Since BH2 reaches the bed and shows good data quality, which we attribute to the nearly vertical shape of the borehole, most of the results presented in this paper are inferred solely from BH2.

Deformation data during the first few months are affected by insufficient mechanical coupling between the tiltmeters and
the ice. We illustrate this with some representative unfiltered tilt curves in panels (c) to (f) of Figure 1. Most sensors show an early period of noisy signals due to poor coupling to the ice, followed by a much longer period of stable tilt evolution. The time at which the transition between noisy and steady tilt change occurs varies from sensor to sensor, from about two months for deeper sensors, where borehole closure from creep is faster, to about six months for shallower sensors, where borehole closure from creep is longer. Most tiltmeters ended up being fully coupled before February 15, 2020, 5 months after installation (see
Supporting Information S1), when we start our analysis. Certain sensors, like BH2#1 or BH2#14, attained their minimum tilt after September 2019, indicating they were initially tilting against the flow. Others, like BH2#8, reached their minimum tilt at, or just after, installation. Other sensors, such as BH2#12, show more erratic behavior over a longer period of time. Note the difference in tilt magnitude at different depths in Figure 1: BH2#1 has a total tilt change of about 60°, BH2#8 of about 25°, while BH#14's tilt change is less than 5°. We removed high frequency noise by smoothing the tilt data with an exponential
filter with a one-day time window. This filtering mainly affects short-term observations and has a negligible effect on long-term (i.e. weekly or monthly averages) analysis. The tilt timeseries of each sensor of the three boreholes BH2, BH3 and BH4 can be found in Supporting Information S1.

## 2.3 Surface motion instrumentation

Five Global Positioning System (GPS) stations were deployed in the ablation zone of the Argentière glacier in February
2019, with an additional seven stations installed in February 2020. This GPS network (green dots in Figure 1) covers the





borehole sites, with station ARG1 as close as 29 m to BH2. The GPS antennas are mounted on aluminum poles anchored up to 6 m deep in the ice. The distance between neighboring survey stations ranges from 50 to 200 m. Regular field visits (monthly at most) ensure the upright position of the antenna poles and continuous power supply. We employ multi-frequency Leica GR25 receivers and Leica AS10 antennas, which continuously record GPS signals at a 1 Hz sampling interval. The

raw GPS data is decimated to 30-second intervals and converted into 24-hour-long RINEX (Receiver INdependent EXchange) format files. These files are processed using a static approach with a double difference processing technique and ionosphere-free linear combination (LC) phase observables (Bock et al., 1986), incorporated in the high-precision geodetic software package GAMIT/GLOBK (Herring et al., 2018). This strategy effectively eliminates phase biases from satellite and receiver clocks. In our static processing, we calculate a single position over 24 hours of data, although the Argentière glacier moves by

more than 10 cm per day. This methodology cumulates all available satellite data whose number is reduced due to the deep valley situation of the network. It determines the position over the processing period that explains best all of the observations while averaging out multi-path interference. Daily position time series are converted into horizontal velocity time series by subtracting successive positions over a 24 hours interval and dividing by the one-day time interval. In case the time interval of two successive positions is longer than one day (due to data gaps from missing measurements or removed outliers), no

velocity is calculated. This avoids biased velocity estimates integrating position offsets unrelated to the glacier motion (e.g. re-installation of the antenna mast). We empirically determined a velocity error of 0.9 m/yr using a stationary GPS station located on the bedrock approximately 500 meters away from the survey network, which is exposed to a similar multi-path scattering environment as the stations on the glacier.

## 3 Methods

### 3.1 Calculating internal deformation rates from tilt-meter observations

We use a three dimensional reference system with $x$ the main along flow direction, and $z$ the upwards vertical with origin at the surface (see panels (a) and (b) of Figure 1). Velocities in $x$, $y$ and $z$ directions are denoted by $u$, $v$ and $w$, respectively. Assuming that the temporal evolution of the tilt $\theta$ occurs entirely in the along flow direction, the changes in $\theta$ are controlled by the horizontal shear strain $du/dz$ and the compressive/extensive strain $du/dx$ and $dw/dz$ (Keller and Blatter, 2012). To detect

the potential effect of compressive/extensive strain, we fit the tilt curve recorded in BH2 at each inclinometer with the Keller and Blatter (2012) analytical model (see Supporting Information S2). We find that the best fit with the data can always be obtained by neglecting $du/dx$ and $dw/dz$ apart for the two deepest sensors (BH2#1 and BH2#2) where a better fit is obtained with non-zero compressive strain. Indeed, neglecting here the compressive strain would lead to an overestimation of $du/dz$ by about 30% and 20% at the sensor BH2#1 and BH2#2 respectively. The presence of significant compressive strain near the

glacier bed is likely related to a local effect of bed roughness that is negligible elsewhere, which is confirmed by the numerical model (see Section 4.2). This is also consistent with the local evaluation of strain over a rough bed done in Section 5.1, which suggests that the tilt curves are significantly affected by horizontal compressive/extensive strain $du/dx$ below 220 m depth in a




layer we will refer to hereafter as the boundary layer. The derived $du/dz$ in this layer are therefore likely to be strongly biased and will be ignored in the ice rheology interpretation.

With the hypothesis that $du/dz$ dominates the flow gradient outside of the boundary layer, the internal deformation rate $du/dz$ from the temporal evolution of the tilt $\theta$ is computed as (Lüthi et al., 2002; Ryser et al., 2014; Doyle et al., 2018; Maier et al., 2019),

$$\frac{du}{dz} = \frac{1}{dt}\frac{dx}{dz} \approx \frac{1}{\Delta t}\Delta\tan(\theta), \tag{1}$$

where $\Delta t$ is a given time period and $\Delta\tan(\theta)$ is the change in the tangent of tilt during that time period. In our particular
implementation, we calculate the least squares linear approximation of $\tan(\theta)$ as a function of time during each $\Delta t$ period (set to one day), such that the regression coefficient gives directly $du/dz$. The estimated maximum uncertainty for the daily averaged deformation rate is, for most of the sensors, lower than 0.1 a$^{-1}$, more details can be found in the Supporting Information S3.

### 3.2 Computation of surface, internal and basal velocities timeseries

As it is not always possible to directly interpolate surface velocity at the borehole location from the GPS network, we construct a continuous timeseries of the surface velocity at the GPS station close to the boreholes by filling gaps using a linear model traditionally used to construct continuous surface mass balances from sparse data on alpine glaciers (Lliboutry, 1974; Vincent et al., 2017). In this linear model we assume similar temporal variability in surface velocity across stations, such that surface velocity at each GPS station $i$ can be expressed as

$u_{si}(t) = \alpha_i + \beta(t), \tag{2}$

with $\alpha_i$ the average surface velocity at the station $i$ over 2020, and $\beta(t)$ the temporal variability assumed identical for all stations and satisfying $\sum\beta(t) = 0$. We first solve the system of equations by finding the values of $\alpha_i$ and $\beta(t)$ that best approximate the observations while keeping $\sum\beta(t) = 0$. We then compute the residuals between modeled and observed velocity and their standard deviation $s_{res}$, and classify as outliers all observations with residuals greater than $3s_{res}$. We finally solve
the system again without these outliers to obtain the final $\alpha_i$ and $\beta(t)$. The residuals of the reconstructed velocities follow a normal distribution centered close to 0 which validate the initial assumption (Supporting Information S4). The surface velocity timeseries $u_s = \alpha_{ARG1} + \beta(t)$ ($s_{res}$ = 3.2 m a$^{-1}$) is thus used to fill the gaps at the ARG1 station, which lies the closest to BH2 as seen in Figure 1.

  The internal velocities are computed by integrating the deformation rate over depth,

$u_d(z,t) = \int\limits_{z_{bed}}^{z} \frac{du}{dz}(z,t)dz \tag{3}$

where $z_{bed}$ is the bedrock elevation (m). Finally, the basal velocity $u_b(t)$ is computed as the difference between the reconstructed surface velocity $u_s(t)$ and the integrated deformation rate over the whole ice thickness $u_d(t)$. The timeseries are



computed at daily resolution from daily averaged deformation rate. We also compute the daily average of the sliding velocity

at the cavitometer with a precision of about 3.5 m a-1 and the subglacial discharge with a threshold of approximately 10 m3

s-1 (Vincent and Moreau, 2016).

### 3.3   Quantifying Glen's flow law parameters

Assessing the ice rheology requires confronting the deformation measurements to the stress tensor $\tau_{ij}$ that need to be evaluated.

We use the three-dimensional full-Stokes finite-element model Elmer/Ice (Gagliardini et al., 2013) to solve for conservation of

momentum equation under a given glacier geometry and ice rheology. The glacier geometry is prescribed based on measured

bedrock topography and surface topography derived from Pleiades satellite imagery on August 25, 2019 (Beraud et al., 2022).

The ice rheology is given by Glen's flow law:

$$\dot{\varepsilon}_{ij} = A\tau_e^{n-1}\tau_{ij}, \tag{4}$$

where $\dot{\varepsilon}_{ij}$ and $\tau_{ij}$ are respectively the components of the strain rate ($a^{-1}$) and deviatoric stress (MPa) tensors, $A$ is the

creep factor ($\text{MPa}^{-n}\ a^{-1}$), $\tau_e = \sqrt{\frac{1}{2}\tau_{ij}\tau_{ij}}$ the effective stress (MPa) and $n$ the Glen's exponent. We assume a stress-free upper

surface boundary condition and a basal boundary condition given by a Weertman friction law (Weertman, 1957),

$$A_s\tau_b^m = u_b, \tag{5}$$

where $\tau_b$ is the basal shear stress (MPa), $m$ an exponent taken equal to 3 (Gilbert et al., 2023), $A_s$ is the sliding coefficient at

the bed (m $a^{-1}$ $\text{MPa}^{-m}$), and $u_b$ the sliding velocity (m $a^{-1}$). We use values of $A_s$ for m=3 taken from Gilbert et al. (2023)

inferred from surface velocity inversion. To avoid stress anomaly produced by uncertainty in the ice thickness, we relax the

surface topography for 1 year, using surface mass balance forcing from Gilbert et al. (2023), before extracting the deformation

rate and stress tensor from the model.

We run several simulations to test the sensitivity of the deformation rate profile to different values of $A$ and $n$. We run a

set of simulations with $n = 3, 4, 5$ and constant and uniform creep factor $A$. The value of $A$ for each $n$ is chosen such that

the numerically calculated total deformation velocity at the location of BH2 matches the observations. We run another set of

simulation with $n = 3$ and depth-dependent creep factor $A = A(z)$ such that the computed deformation rate $du/dz$ match the

observations. To find the creep factor as a function of depth, we infer $A$ by using Glen's law (Eqn. (4)) knowing the observed

mean $du/dz$ at BH2 and the stress tensor computed from the Elmer/Ice simulation such that

$$A(z) = \frac{1}{2}\frac{du}{dz}\tau_{E,num}^{-2}\tau_{xz,num}^{-1}. \tag{6}$$

We then approximate $A(z)$ by a piece wise linear function $A_{fit}(z)$ to obtain a depth-dependent viscosity. Given that changing

the creep factor slightly modifies the overall stress balance, we run the numerical model repeatedly, updating at each iteration





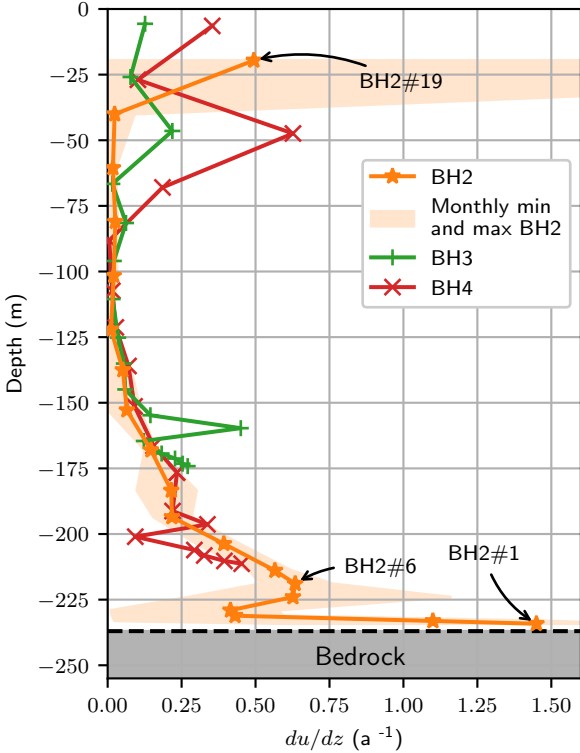

**Figure 2.** deformation rate profiles with monthly minima and maxima at BH2, BH3 and BH4. The continuous lines show the average measured deformation rate profile at each borehole for the period between the 15th February and the 15th October 2020, and the shadowed region the range between monthly averaged minima and maxima deformation rate values (shown only for BH2). Every symbol represents a tiltmeter.

the $A_{fit}(z)$ inferred with the numerical solution of the previous iteration, until the modeled stress field converges. The depth-dependent creep factor A(z) determined at BH2 is applied uniformly over the entire domain by normalizing it with depth and applying this normalization everywhere.

# 4   Results

## 4.1   Observed mean deformation rate profiles


deformation rate profiles computed from tilt measurements (see methods in Section 3.1) at BH2, BH3 and BH4 and averaged between the 15th February and the 15th October 2020 are shown in Figure 2. The orange shaded region shows the range associated with monthly-averaged deformation rate $du/dz$ for BH2. The retrieved profiles show similar depth-increasing deformation rate, but only BH2 reached the bed (see Figure 1b and Table 1). BH2 also shows generally lower measurement noise likely due





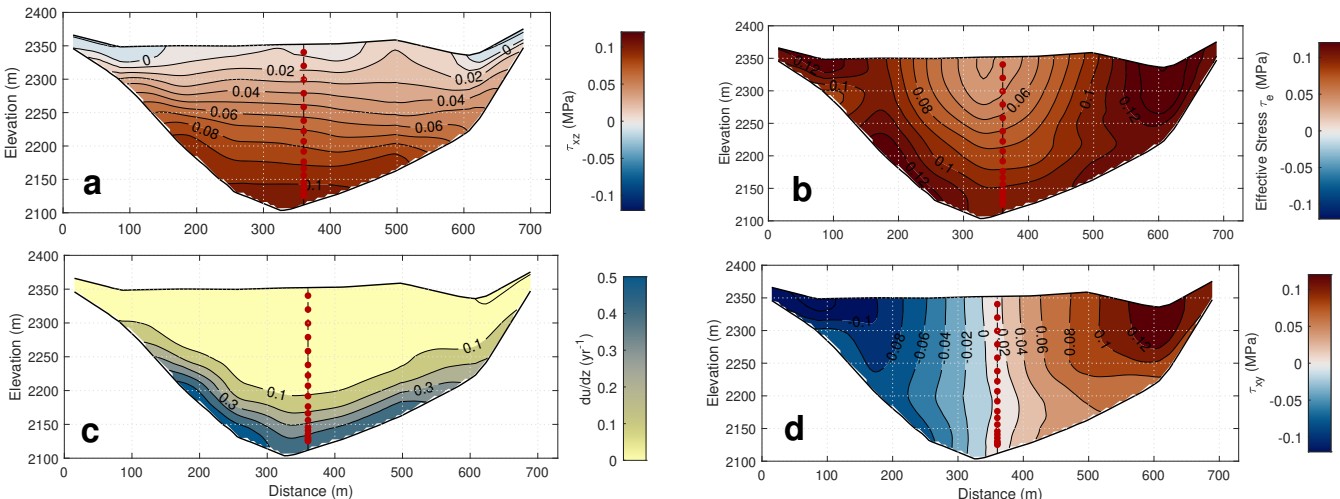

**Figure 3.** Modeled shear stress $\tau_{xz}$ (a), effective stress $\tau_e$ (b), deformation rate $du/dz$ (c) and shear stress $\tau_{xy}$ (d) along a transversal cross section at BH2 location. The inclinometers from BH2 are shown as red dots.

to better ice coupling (see Supporting Information S1). For both of these reasons, from now on we focus on the deformation rate profile recorded at BH2 only, which we divide in three parts.

The upper part of the deformation rate profile, spanning the uppermost 120 m, has a small shear deformation rate ($\approx 0.02\ a^{-1}$). BH2#19 shows very noisy records and is thus disregarded from our analysis (see Supporting Information S1). The middle part, from -120 m until -219 m (BH2#6), is characterised by much higher deformation rates increasing non-linearly towards the bed,

from less than 0.02 a$^{-1}$ at -120 m to a local maximum of 0.63 a$^{-1}$ at -219 m. Below -219 m starts the lowest part, which we refer to as the boundary layer. This part includes a 40% decrease in deformation rates over the first 10 meters below -219 m, followed by a more than threefold increase in $du/dz$ between -230 m and -235 m, up to almost 1.5 a$^{-1}$ near the bed at -235 m, which corresponds to the highest deformation rate of the profile.

## 4.2  Ice flow model

Ice flow at the BH2 site is found to be mainly dominated by along flow shear $\tau_{xz}$ and lateral shear $\tau_{xy}$ (Figure 3). The lateral shear can reach significant values away of the central line and greatly affect the effective stress $\tau_e$ and thus deformation rate on the side of the glacier (Figure 3). However, $\tau_{xy}$ remains very small compared to the shear along the flow $\tau_{xz}$ at the drilling site, making our measurement principally influenced by $\tau_{xz}$. Other components of the stress tensor have been found to be small compared to $\tau_{xz}$, except near the surface where they can be of similar amplitude (see Supporting Information S5).

Modeled flow gradient tensor components $du/dx$ and $dw/dz$ are very small (see Supporting Information S5) which validates the use of Eq. (1) to calculate $du/dz$ (see Section 3.1). We also found that the stress tensor is rather insensitive to the choice of rheological parameters and is mainly controlled by the glacier geometry (Supporting Information S5). Comparing our result with a simplified plane-strain model, commonly referred to as the Shallow Ice Approximation (SIA), we show that the





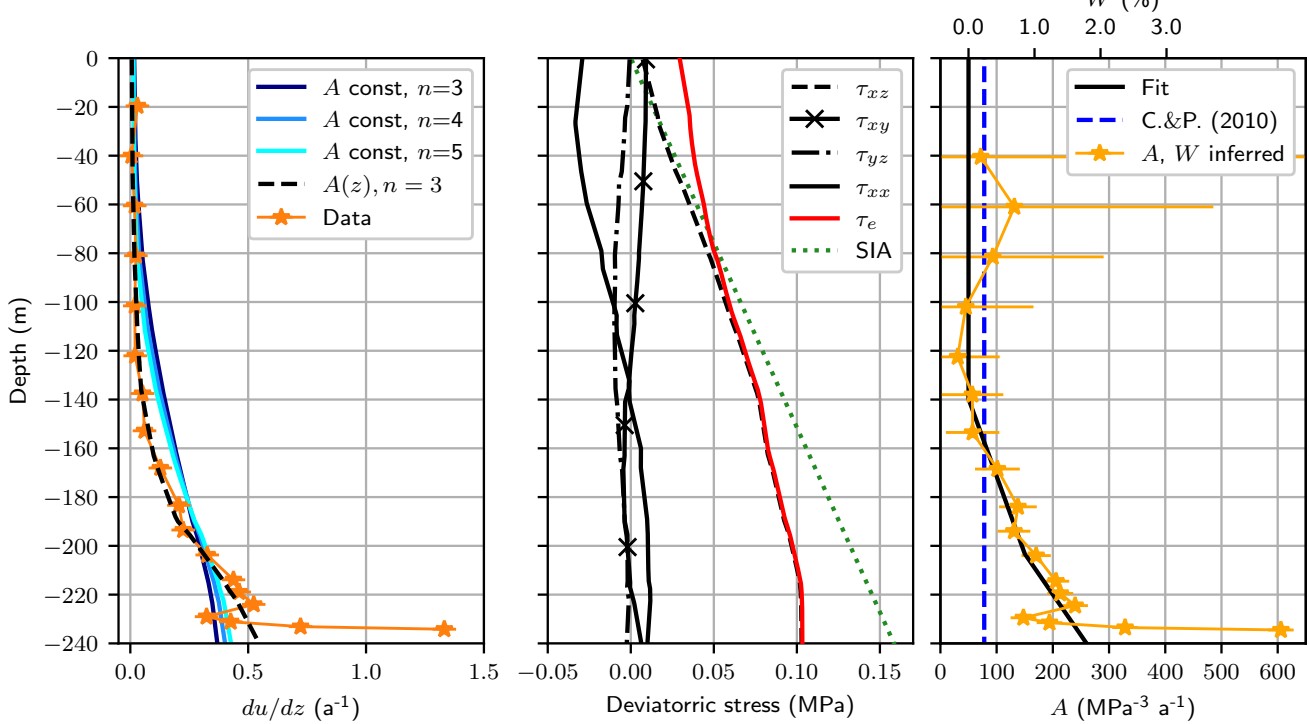

**Figure 4.** (a) Observed and modeled deformation profile at BH2 with uniform creep factor and $n = 3$, $n = 4$, and $n = 5$, and with depth-variable creep factor $A = A(z)$ and $n = 3$. (b) Vertical profile of different modeled stress components (compressive is negative). The Shallow Ice Approximation (SIA) solution without shape factor is given for comparison (green dotted line). (c) Inferred creep factor using measured deformation rate and modeled effective stress at BH2 from Eq. (6) (bottom horizontal axis). The corresponding water content according to Duval (1977) is shown by the top horizontal axis. The blue dashed line marks the value of $A$ proposed by Cuffey and Paterson (2010).

SIA would overestimate the shear stress (see Figure 4b), leading to an incorrect quantification of the rheological parameters.
Introducing a shape factor of $f = 0.646$ into the SIA formulation (Nye, 1965) would give however a fairly good result. This shape factor is an appropriate value for a parabolic valley with a half-width to thickness ratio of 2 (Nye, 1965), a reasonable approximation of the Argentière glacier cross-section at the study site (see Supporting Information S6).

To minimize the potential effect of temporal changes in basal friction on the stress field (Hooke et al., 1992; Willis et al., 2003), we base our analysis of ice rheology on the averaged deformation rate profile during the last month of the timeseries
(October $1^{st}$ to 31, 2020) when the basal friction is the least affected by subglacial hydrology and the inclinometers are well coupled to the ice. We extract the deformation rate from the three-dimensional ice flow model at BH2 location and find that simulations with constant creep factor $A$ and varying Glen's law exponent $n$ yield deformation rate profiles exhibiting much lower non-linearity with depth than observed regardless of the flow exponent ($n = 3$, $n = 4$ and $n = 5$, see Figure 4a). Constant creep factor is thus not able to explain the observed profile regardless the value of $n$ that is poorly constrained by the data. To





be consistent with the commonly used value of $n$ and infer a relevant creep factor for ice flow modeling in general, we thus assume that $n = 3$ and quantify $A$ following the method described in Section 3.1 (Figure 4c).

Using the linear piece wise approximation of $A$ (black line in Figure 4c), the simulation provides a good match with observations (Figure 4a), apart from the deformation rates recovered in the boundary layer which are poorly reproduced by the numerical model. This is likely because the resolution of the bed topography is too low to capture the stress field generated by 285 the unresolved local bed roughness and also because the data themselves do not reflect the real $du/dz$ due to significant compressive/extensive strain that are neglected in the boundary layer (see Section 3.1). We consider that the values of $A$ inferred in the boundary layer are not relevant for the same reason. In Figure 4c, we see that the inferred creep factor $A(z)$ in the upper half of the glacier is compatible with the value proposed by Cuffey and Paterson (2010) for temperate ice but increases from $-140$ m down to the top of the boundary layer up to a factor $\approx 4$. We discuss this depth-increasing creep factor in the Section 290 5.2.

## 4.3 Seasonal evolution of velocity

On the seasonal timescale, the surface velocity shows an annual cycle with a gradual increase between December and May, a period of stagnation until mid-September and a decrease until December (Figure 5). The period of stagnation coincides with the melting period as shown by the observed high discharges. Surprisingly, the deformation velocity shows a seasonal 295 variation with similar phase and amplitude to that of surface velocity, suggesting that the basal velocity remains roughly constant throughout the year. This is in contrast to in-situ observations of basal velocity at the cavitometer, where sliding velocities show a strong seasonal variability materialized by highest sliding speeds in July and lowest in February/March (Figure 5b, see also Gimbert et al. (2021a)). These results are further discussed in section 5.3.

On shorter timescales, the relationships between surface, deformation, and sliding velocities are different from those ob-300 served on seasonal timescales. Peaks in surface velocity that occur at the beginning of the melt season coincide with a decrease in deformation velocity and a strong increase in basal sliding (see highlighted as vertical dashed lines in Figure 5). These peaks are also visible at the cavitometer, particularly for the May events. At the end of the melt season (October), peaks of surface velocities are also mainly explained by changes in sliding speed but not necessarily coincide with a decrease of deformation velocity as observed for the early melt season peaks.

To better understand the origin of the deformation velocity changes, we investigate whether the temporal variability in $u_d$ is due to changes in $du/dz$ in a specific section of the ice column such as for example in the boundary layer. We define $s_n$ as the standard deviation of the monthly averaged $du/dz$ profile normalized by $du/dz$ averaged between 15th February and 15th October 2020. We show that $s_n$ is consistently about 20% with no indication of a preferential localisation of the strain-rate variability in a specific section of the ice column (Figure 6b). A more detailed analysis of $du/dz$ variation at each sensor shows 310 that most of them exhibit a similar seasonal cycle with maximum deformation in summer (see Supplementary Information S7). Only four sensors (BH2#6,11,12,14) show a different behavior with different phase (#6,12,14) or amplitude (#11) (Figure S17). The noise level for #12 and #14 is particularly high, and #6 may be influenced by the local stress field in the boundary layer



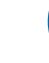 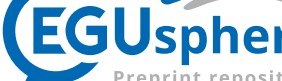

**Figure 5.** Daily timeseries of subglacial discharge (a) and velocities (b) at the Argentière Glacier. The panel (b) shows the surface velocity computed with the linear model for the GPS station ARG1 $u_s$, the deformation velocity at BH2 $u_d$, the inferred basal velocity at BH2 $u_b$ and the sliding velocity at the cavitometer $u_{cav}$. The horizontal dashed lines correspond to the average velocities over the studied period. In the shaded area, an estimation of the sliding velocity (black line) is shown assuming a linear trend in the deformation velocity (blue line). The vertical dashed line show speed up events that are referred to in the text.

(see section 5.1). However, there is no obvious explanation why #11 shows so little variation of du/dz while its average value is coherent with the du/dz depth profile (Figure 4).




**Figure 6.** (a) Monthly and yearly $du/dz$ profile at four periods (see legend). (b) Standard deviation of the monthly averaged $du/dz$ profile normalized by the $du/dz$ averaged between 15th February and 15th October 2020. (c) Monthly and yearly horizontal velocity $u_h$ profile at four periods (see legend).

## 5 Discussion

### 5.1 Identification and interpretation of the boundary layer

The shape of the deformation rate profile and the retrieved values of $du/dz$ close to the bed suggest a boundary layer due to sliding over a bump as explored by Maier et al. (2019). We qualitatively explored the viability of this explanation with a



| Study | Glacier | Area | Avg. $W$ (%) | Method | Notes |
|---|---|---|---|---|---|
| Joubert (1963) | Vallé Blanche (France) | Accum. | 0.15 - 0.1 | Calorimeter | Only surveyed upper part of the column |
| Lliboutry (1971) | Vallée Blanche (France) | Accum. Ablat. | 0 - 0.6 0 - 1.7 | Calorimeter | No details on the spatial distribution |
| Vallon et al. (1976) | Vallée Blanche (France) | Accum. | 0.32 - 1.31 | Calorimeter | Depth increasing $W$, drops at the bed |
| Zryd (1991) | Findelengletscher (Switzerland) | - | $0.5 - 1.5$ | Calorimeter | Basal ice |
| Cohen (2000) | Engabreen (Norway) | - | 1 (cloudy ice) $> 2$ (with sediments) | Calorimeter | Basal ice |
| Murray et al. (2000) | Falljökull (Iceland) | Ablat. | 0 - 3.3 | GPR | Depth increasing $W$, drops at the bed |
| Benjumea et al. (2003) | Johnsons Glacier (Antarctica) | Accum. | 0.6 - 2.3 | GPR and seismics | Depth increasing $W$, drops at the bed |
| Hubbard et al. (2003) | Tsanfleuron (Switzerland) | Both | $W_0$ - $10.7 W_0$ | Ion concentration | Depth increasing $W$. $W$ is quantified relatively to its unknown upper layer value $W_0$ |
| Murray et al. (2007) | Tsanfleuron (Switzerland) | Ablat. | 1.18 - 3.8 | GPR | Depth increasing $W$. |
| M. Lüthi (personal communication) | Argentière Glacier (France) | Ablat. | 2 | Calorimeter | Basal ice |

**Table 2.** Synthesis of water content measurements made in temperate glaciers, with emphasis on the studied area of the glacier and the in-depth distribution.

simulation of tilt evolution close to the bed using the deformation rates provided by Gudmundsson (1997) and the model of tilt
evolution in a given velocity field provided in Gudmundsson et al. (1999), which we explain in the Supporting Information S8. In Figure S18, we show that significant compressive or extensive horizontal strain-rates $du/dx$ localise close to the bed and dominate the flow gradient in a boundary layer with a thickness equal to $\approx 6$ times the amplitude of the bedrock bump. We find that under the hypothesis of Eq. (1) and using synthetic tilt timeseries produced from the estimated strain-rates, the inferred apparent $du/dz$ would have a zigzag shape similar to the one observed in Figure 2. This suggests that the zigzag shape in the
data is an artifact resulting from neglecting $du/dx$ and not a real variation in $du/dz$. The comparison between data and the inferred apparent $du/dz$ using Gudmundsson (1997) and Gudmundsson et al. (1999) models suggest that the drill reach the summit of a bed rock bump of $\approx 2$ m amplitude.



## 5.2 Interpretation of depth-increasing creep factor

Since Argentière Glacier is fully temperate, depth-increase of the creep factor may be due to an increase in interstitial water
content $W$ with depth (Duval, 1977; Adams et al., 2021). We test this hypothesis using the formula proposed by (Duval, 1977)
to convert the estimated depth-increase in creep factor into an associated increase in water content. Adapting the formulation
from Duval (1977) by considering $A = 50$ MPa$^{-3}$ a$^{-1}$ when $W = 0$ as a reference value, i.e., assuming no water content in the
upper half of the glacier, we obtain

$$W = \frac{1}{2.34} \left( \frac{A}{50} - 1 \right),$$
(7)

with $W$ in % and $A$ in MPa$^{-3}$ a$^{-1}$. The inferred water content values are given in the top horizontal axis of Figure 4c.
Discarding the negative values as artifacts of our chosen parameterization of Duval's model, we see that the expected water
content above -219 m ranges between 0 and 1.3%, increasing down to the bed. Below this depth, the deformation profile is likely
influenced by enhanced stress and underestimated velocity gradients due to local basal roughness and cannot be interpreted
by enhanced creep factor (see Section 5.1). These values of $W$, and this type of spatial distribution, are comparable to those
observed in temperate ice (see Table 2). The depth-increasing creep factor is thus compatible with the effect of depth increasing
water content.

Our finding of increased water content with depth could appear to be in contradiction with that of of Lliboutry and Duval
(1985), who report no relationship of $W$ with depth through analysis of an ice core obtained in the Argentière Glacier in
a location close to our boreholes (Hantz and Lliboutry, 1983). However, the engineers and researchers that performed the
measurements deem them untrustworthy as a result of water content being primarily correlated with the air temperature under
which the measurements were made at the glacier surface, with higher temperatures increasing ice core melt and thus artificially
increased water content (Michel Vallon, personnal communication). For these reasons we do not include Lliboutry and Duval
(1985) is not included in Table 2 although it previously appeared in well cited compilations of observations of interstitial
water content (e.g. Pettersson et al., 2004; Cuffey and Paterson, 2010). We also note that the absolute values of water content
as inferred using the relationship proposed by Duval (1977) may be associated with high uncertainty, since this empirical
relationship was obtained in the laboratory by shearing temperate ice in tertiary creep with water contents up to 0.8% and
has not been validated for higher values of water content. Recently, Adams et al. (2021) found in similar experiments that ice
viscosity under secondary creep is nonsensitive to water content for $W > 0.6$%, but we do not expect these results to apply in
the present case since tertiary creep is expected under the high cumulative deformations of Argentière Glacier (Lliboutry and
Duval, 1985; Budd and Jacka, 1989).

Although water content is our identified best candidate to explain the increase in creep factor with depth, we note that
other factors could be at play, such as depth-decreasing grain size or changes in ice anisotropy (Cuffey and Paterson, 2010;
Montagnat and Duval, 2004). However, Vallon et al. (1976) reported no discernible change in grain size except at the bed in
the accumulation zone of La Mer de Glace, a glacier close to the Argentière Glacier. The role of anisotropy would need to be



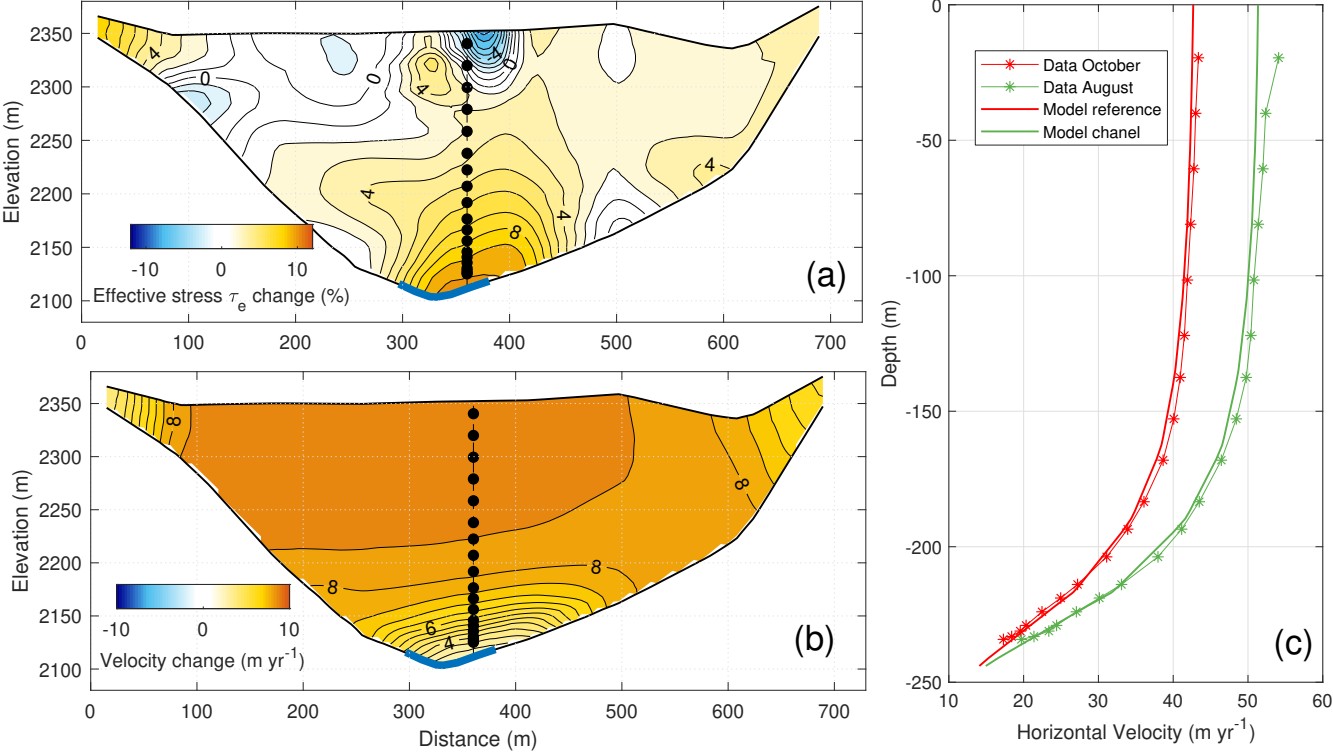

**Figure 7.** Modeling change in effective stress (a) and horizontal velocity (b) due to non-uniform variation of basal friction. We assume that, during melting period, friction increase in the central part (blue thick line on panels (a) and (b)) and friction decrease elsewhere. The inclinometers of BH2 are shown as black dots in (a) and (b). Panel (c) shows modeled and observed velocity profile at BH2 in August and October.

specifically analyzed by quantifying the evolution of ice textures with depth at the measurement site to be able to discard or not its influence on ice viscosity.

### 5.3   Temporal changes in deformation velocity

We show that the deformation velocity varies seasonally by $\approx 30\%$ (Figure 5), which, using Glen's flow law with $n = 3$, implies either a stress change of $\approx 9\%$, either a change in the creep factor of $\approx 30\%$. The seasonal variability of the creep

factor value could be explained by a seasonal change in water content of $\approx 60\%$ uniformly over the entire ice column, which seems unrealistic, since the increased ice deformation in summer would produce an excess of water content of only $\approx 0.17\%$ due to increased strain heating. Also, the stress changes cannot be explained by the seasonal evolution of ice thickness, which is maximum in May and decreases until October (Vincent et al., 2022) and thus out of phase with the observed deformation rate. Furthermore, the amplitude of the thickness change at the BH2 site is about 6 m (Vincent et al., 2022), which represents only

a 2% stress change. The most plausible hypothesis to explain the seasonal variability in deformation rate observed at BH2 is




through changes in stress distribution due to non-uniform spatial changes in basal drag during the melting season as previously suggested in Hooke et al. (1992) and Willis et al. (2003). The loss of drag in response to meltwater input in areas surrounding the observation site can locally increase stress through stress transfer in the form of lateral shear. This could be caused by differences in subglacial hydrological conditions between the glacier centerline and its sides. The lack of change in basal

velocity during summer at BH2 (Figure 5), while deformation velocity and stress increased, suggests that basal friction also increased in the central part of the glacier to accommodate more stress while remaining at a constant velocity. This could result from the development of efficient drainage in the central part of the glacier, as previously identified from seismic observations by Nanni et al. (2021), which reduces water pressure and thus promotes high friction along the central line, while the sides remain dominated by higher pressure inefficient drainage promoting lower basal friction. To investigate this hypothesis, we

perform a complementary numerical simulation where we increase the friction around a central ≈50 m wide area (assumed to be affected by an efficient drainage) and decrease it elsewhere (see Figures 7a and 7b). We find that, relative to the reference state in October 2020, a decrease of 20% of $A_s$ in the central part combined with an increase of 70% of $A_s$ elsewhere is able to produce the stress change needed to explain the enhanced internal deformation observed in summer (Figure 7c). Such a contrast in $A_s$ (factor 2.1) is compatible with that expected when cavitation occurs or not (Gimbert et al., 2021a; Gilbert et al.,

2022; Maier et al., 2022). This view is also consistent with the findings in Vincent et al. (2022) that bed separation by cavitation increases between January and July with greater amplitude on the glacier margin than on the midline. It leads to an increase in basal sliding during this period, as observed in our data (Figures 5 and 6c). An evolving drag contrast between the center and side of the glacier would also explain why strong peaks in surface velocity are associated with a decrease in deformation velocity during the early melt season. Early water input into the not-yet-developed efficient drainage in the centerline of the

glacier may lead to pressurization of the central channel, reducing drag in the centerline relative to the side of the glacier.

## 6   Conclusions

Using borehole inclinometry techniques, we were able to reconstruct the deformation profile along the central line of the ablation area of Argentière Glacier and its evolution over eight months, including the entire melt season. These observations allow us to quantify the Glen's flow law creep factor by modeling the local stress field using a three dimensional full Stokes

ice flow model. We show that surface values (above 100 m depth) of the creep factor are consistent with the standard value for temperate ice (Cuffey and Paterson, 2010), but increase progressively below this depth up to 200 MPa$^{-3}$ a$^{-1}$. We interpret the depth-increasing creep factor as an associated increase in water content from 0 to 1.3 percent using Duval (1977)'s law, which is a reasonable values given previous in-situ water content measurements from the literature (Table 2). We also find seasonally evolving deformation rates, with higher deformation rate occurring during the melt season. The evolution of the deformation

velocity explains most of the evolution of the surface velocity on a seasonal time scale. We show that the seasonal variability in deformation rate at the study area is due to changes in stress distribution within the ice body in response to the evolving contrast in basal drag between the centerline and the rest of the glacier during the melt season. We interpret the difference in drag at the center of the glacier as the effect of efficient drainage developing at the deepest point of the subglacial valley.



This study demonstrates the significant value in using borehole inclinometry for inferring ice rheology and local changes in
basal friction that cannot be detected through surface velocity observations. The data obtained provides rare insights into how
subglacial hydrology, basal friction, and surface velocity are interconnected.

*Code and data availability.* All data used in this article and the code to process it can be accessed through the Zenodo repository available
at https://doi.org/10.5281/zenodo.11371900

*Author contributions.* JP. Roldán-Blasco processed the tilt data and A. Gilbert designed and performed the numerical simulations. JP. Roldán-
Blasco, A. Gilbert and F. Gimbert wrote the manuscript. L. Piard designed the tiltmeters, directed the field campaign and performed early
analysis on the data. L. Piard, A. Gilbert, F. Gimbert, C. Vincent and O. Gagliardini participated in the inclinometry field campaign and
together with JP. Roldán-Blasco analysed the data, while A. Togaibekov and A. Walpersdorf provided the GPS measurements and velocity
timeseries. All authors contributed to finalizing the manuscript.

*Competing interests.* The authors declare no competing interests.

*Acknowledgements.* The authors would like to thank Bruno Jourdain, Olivier Laarmann, Maël Richard and everybody else that participated
in the 2019 field campaign and the fabrication of the instruments. This work is supported by the French ANR project SAUSSURE (ANR-18-
CE01-0015-01, https://saussure.osug.fr). We also thank Luc Moreau for providing the cavitometer data and Electricité d'Emosson for the
discharge measurements. Glacier surface elevation and precipitation data were acquired in the framework of the GLACIOCLIM program
(https://glacioclim. osug.fr). OG acknowledges support from the project FricFrac funded by the Center for Advanced Study (CAS) at the
Norwegian Academy of Science and Letters during academic year 2023-2024.



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
