# Peer review of "Creep enhancement and sliding in a temperate, hard-bedded alpine glacier"

_EGUsphere, 2024_

## Referee Comment (RC2)

**Creep enhancement and sliding in a temperate, hard-beded alpine glacier**

**Summary**

The manuscript titled "Creep enhancement and sliding in a temperate, hard-beded alpine glacier" by Roldán-Blasco et al. examines the basal sliding and internal ice deformation of an alpine glacier. The authors observe an increasing creep factor with depth, which they attribute primarily to changes in water content. However, they suggest that seasonal changes are more likely due to variations in stress distribution within the glacier, as neither the increase in water content nor the stress change alone fully accounts for their observations.

The study is based on approximately 20 tiltmeter measurements along a borehole over an 8-month period. These data are complemented by glacier flow velocity measurements from the glacier surface and bed. Additionally, the authors employ a modeling approach to better understand stress distribution within the glacier. By connecting the modeling output with the tilt and discharge measurements, they quantify the Glen's flow law creep factor, investigate the water content within the ice relative to depth, and analyze stress distribution within the glacier. Finally, they link their findings to the state and evolution of the subglacial water system.

Overall, this manuscript presents significant observations regarding Glen's flow law creep factor and the seasonal evolution of the subglacial water system. Minor comments for improvement focus mainly on enhancing readability.

Manuela Köpfli

**General questions**

- Would you assume to see similar inclinometer measurements for all boreholes, if they would have worked properly and the installation/drilling would have been perfect?
- I understand that you mainly focus on the tilt measurements. If possible, I would appreciate seeing a figure of the initial and final borehole shape as Figure 1.b but also showing the final borehole shape.
- In Figure 2, BH3 shows this larger value at depth around -160m. Do you think this is real? If so, do you have any explanation for that since your boreholes BH2 and BH3 are very close?
- In Figure 5 you show your data with a daily resolution. Would it be possible to to "zoom in" to the first half of May and have a higher time resolution for velocities and discharge. It would be very interesting to see, if you see a delay between glacier speed up and discharge.

**Comments Text**

Note: I recognized that you used different ways to write a date: 15th of February, 15th February, 15$^{th}$ February. Maybe you choose one format and keep it through the whole manuscript. I just wrote "date" and the corresponding line.

Title: I would use "bedded" instead of "beded"

L.32-35: I would break the sentence into two sentences.

L.94: date

L.139: date

L.201: To me, the 2020 is a bit misleading. I would exchange it with something like "over one year (2020)" or "over eight months (2020)" or even remove the 2020 completely.

L.205: To me, there are no residuals of the reconstructed velocities. As I understand, the residuals are more between the reconstructed and observed velocities

L.214: I don't really understand the word "threshold" in this context.

L.214-215: unit goes over the line and exponent is not really an exponent (for both units: m a-1 and m3 s-1)

L.223-237: I'm a bit confused by $n$ and $m$. If they are not the same, could you add a sentence and describe the difference?

L.228: I would write *m* italic that it is immediately clear that you are not talking about meters (like L. 235)

L.236: Normally you used Eq. instead of Eqn.

L.242: I prefer the italic writing of parameters: *A(z)*

L.246: Maybe change this sentence to: "Averaged deformation rate profiles, computed from tilt measurements (see methods in Section 3.1) at BH2, BH3, and BH4 between February 15 and October 15, 2020, are shown in Figure 2."

L.247: date

L.275: October 1$^{st}$ to 31$^{st}$ (second superscript is missing), date

L.298: Normally, you wrote section starting with a capital letter

L.307: date

L.313: Normally, you wrote section starting with a capital letter and italic du/dz

L.314: Italic du/dz

L.345: For these reasons we do not include Lliboutry and Duval (1985)  in Table 2…

L.364: … either a stress … OR a change…

L.397: use % instead of writing percent

**Comments Figures**

Figure 1: Make sure that all blue dots are visible (especially BH1). Maybe use a different color for the label of the GPS station "ARG1" so that it is immediately clear that this is something different and there is no blue dot behind the green dot. Add to the last sentence in the figure caption, why you start your analysis on 15th of February. date (in the figure caption)

Figure 2: Start figure caption with a capital letter. I would rewrite the figure caption as "Average measured deformation rate profiles  at BH2, BH3 and BH4 including the monthly minima and maxima for BH2.  between the 15th February and the 15th October 2020

 Every symbol represents a tiltmeter." date (in the figure caption)

Figure 4: Labels a), b), c) are missing in the figure. Is in subplot c) W only inferred for the orange markers? Is the assumption for the blue line, C & P (2010), that W is constant. If so, I would change the label of the orange line/markers also to A(z) and only mention the water content W in the figure caption (as you do).

Figure 5: I would use a different color for the discharge vs basal velocity, just for clarity. Maybe you want to mention again, that you convolved/averaged your data on a daily basis and you did the same for the discharge? Because, I would assume some daily variation in the discharge over the melting season.

Figure 6: date (in the figure caption)

Figure 7: It is correct what you are saying in your figure caption but when I first read that, I did not believe that you wanted to say that the friction increases during melting.  To avoid that, I would add that this is due to "efficient drainage" or an "evolved drainage system".

**Comments Supplement**

Figure S9: I would use $\Delta t$ instead of "Delatat" and $du/dz$ instead of dud and decide if Error should have a capital letter or not (title vs. color bar label)

Figure S10: Again, i would prefer  $\Delta t$ instead of "Delatat" and $u_d$ instead of "du"

Figure S11: I'm not sure if you need that many positions after decimal point for the std.

Figure S16: The orange line is hard to see. Also the label for (a) is missing. I would suggest making two figures out of the one figure, just take your figure S16a as one figure and add S16b as S17. This allows you to make S16a bigger and easier to understand. Please check the linewidth of the orange line or change the order you are plotting the lines sot that blue and orange are in the front.

---

## Author Comment (AC1)

**Response to reviewers**

Reviewer comments in black *- Answers in blue italic*

**Reviewer 1 (Dominik Gräff):**

In the present manuscript Roldán-Blasco et al. describe how they use glacier borehole inclinometer measurements to invert for the creep factor dependence on depth. From their measurements they infer interesting results about the seasonal variability of basal sliding and provide conclusive explanations for their observations. I evaluate the presented work as a thoroughly carried out analysis that is documented in detail and without any doubt relevant for the glaciological community. The strength of the manuscript lies, in my eyes, in combining field experimental data with realistic ice-flow modeling to learn about in-situ glacier ice viscosity, and to deduce reasonable physical mechanisms to explain their data. The weakness of the manuscript lies, in my opinion, in the long and comprehensive presentation of the study, where I am missing the conciseness.

I recommend the present manuscript for publication in The Cryosphere, 1) because I evaluate the analysis of this study to be done thoroughly and 2) due to the relevance of the results. Therefore, I only make very minor comments and suggestions below which might improve the comprehensibility.

 Best regards,

Dominik Gräff

*We would like to thank the reviewer for his encouraging comments about his interest in our work and for his suggestions, all of which have been incorporated into the manuscript and have improved its quality.*

**General Comments:**

My foremost criticism of this manuscript links to my personal preference for short papers. This manuscript is not extraordinarily long for The Cryosphere. However, in my opinion the authors should try to focus more on the information that is important for the reader to get an overview over the experiment, understand the applied methods, and to retrace the derived conclusions. The present manuscript fulfills this, but it draws the attention of the reader repeatedly to details that are distracting and that make the flow of reading in some way humpy and hilly.

*We have significantly shortened the instrumentation and methods sections and tried to remove all non-essential details throughout the text. We believe that the revised manuscript is easier to read and less distracting from the main points.*

Three examples:

I'm very interested in instrumentation and even more in hot water drilling. However, I think the two paragraphs between L94-118 could be shortened for the main text by at least 75%. You might lose many readers by talking about the communication protocol that the tiltmeters used and it blocks the main massage of your study by entangling the reader in details. The same is in my opinion true for several paragraphs.

*We reduced the length of these two paragraphs by 45% which makes them less likely to lose the reader while keeping some details on the sensors for those who are interested in the technique we used here. We also shortened most paragraphs of the instrumentation and methods sections in the revised manuscript. Overall, the "Field site and instrumentation" section is also 45% shorter in the revised manuscript.*

Throughout the language of the manuscript seems convolved to me. L162/163: 'Daily position time series are converted into horizontal velocity time series by subtracting successive positions over a 24 hours interval and dividing by the one-day time interval.' could be phrased much simpler as 'From the GNSS locations, we calculated daily horizontal velocities.' without losing much of the information.

*We simplified this sentence in the revised manuscript and also shortened most of the paragraphs in the instrumentation and methods sections.*

Often additional information given is simply clear for the scientific reader: 'In case the time interval of two successive positions is longer than one day (due to data gaps from missing measurements or removed outliers), no velocity is calculated. This avoids biased velocity estimates integrating position offsets unrelated to the glacier motion (e.g. re-installation of the antenna mast).' This is like if you were explaining to a mechanic with which hand you are tightening a screw. Again, my personal opinion.

*Yes, we agree, such sentences have been removed in the revised manuscript.*

A scientific comment: You state in L364/365 that for explaining your measured seasonal variations in deformation velocity a change in water content of 60% would be needed and that this seems unrealistic based on strain heating. But why would you rule out that meltwater is pushed into micro-cracks increasing the interstitial water content. My colleagues and I measured at Rhonegletscher a daily increase of 90-180% of micro-scale water content in the glacier ice (Gajek, W. et al. Diurnal expansion and contraction of englacial fracture networks revealed by seismic shear wave splitting. Commun Earth Environ 2, 209 (2021). https://doi.org/10.1038/s43247-021-00279-4). If I understand it correctly, this effect could also explain the seasonal deformation velocity variations.

*Yes, we agree that it may not be justified to completely rule out this possibility, although we think that a change in water content is less likely to explain the change in deformation rate because subglacial water pressure actually decreases during the summer (according to water pressure measurements in BH2). It is also not clear how water content through englacial fracture actually affects ice rheology, for which the effect of water content has been shown for homogeneous interstitial water at the grain boundary. Finally, this change in water content must be uniform throughout the ice column, which may not be the case if the hydraulic forcing comes from surface melting and subglacial water pressure. Nevertheless, we have added a sentence in the revised manuscript to open up the possibility that seasonal changes in water content could explain our observation, and added the reference mentioned by the reviewer (lines 335-340). We also thank the reviewer for pointing out this relevant reference to our study.*

There are a bunch of typos throughout the manuscript or funny plotting issues (eg. Fig.6 180m depth label), that I leave for typesetting. Sometimes units are missing, which is particularly important talking about angles as the quantity could be in radians or degree. Overall, just cosmetics that can be fixed by thoroughly reading through the PDF again.

*We have verified that all units are now specified and corrected the typos.*

 Nice to see such a comprehensive data and code repository on Zenodo!

*Thanks, we updated the Zenodo repository with the modified figures of the revised manuscript.*

 As a last general comment, I would have loved to see a figure showing final borehole shapes like Fig.1b). Maybe even borehole shapes over time. The reason is, that this is in the end what the tiltmeters are measuring. They are located at a certain depth, and they record the inclination of the borehole at that given depth. This information would directly be shown in a depth over horizontal distance plot. I know this manuscript works a lot with the derivatives (eg. u), or even second order derivatives (du/dz). But derivatives intrinsically emphasize high frequency noise and pose some danger when

integrating due to drift. I was relieved, when I came to Fig.6c, but then a bit sad that there were no data points at 20m and 0m and when I saw that the horizontal scale was in units of velocity. That's all fine, but what you are measuring is a location at the surface (x,y,z) and inclinations at depth (theta). The shape would look identical, but there is this subtle difference in plotting recorded quantities and derived ones.

*We agree with the reviewer that it would be an interesting plot to do. However, the tiltmeters are not necessarily well aligned with the borehole and their rate of change rather than their absolute value is a more reliable and valuable information as we attempt to quantify deformation rate and not cumulative deformation. The fact that tiltmeters are not well aligned with the borehole is illustrated by the fact that a lot of tilt measurements go through a minimum value before rising again. This is not an issue when focusing on the rate of change whereas the absolute value is not meaningful. The noise level is reduced by averaging the rate of change which is equivalent to cumulating the total tilt change. Nevertheless, in the revised manuscript, we added the final shape of borehole 2 in Figure 1 for information even though it is not used in the analysis. See also the figure below.*

[Figure]

*Figure R1 - Initial (october 2019) and final (october 2020) shape of borehole 2.*

**Specific Comments:**

In the following, I list some comments and questions that might be helpful to improve the quality of the manuscript directly referring to line numbers of the manuscript:

Abstract:

Nicely written, concise and L9-11 stimulates to read.

1 Introduction:

L24: I'd say: Ice deformation is 'commonly assumed' to follow Glen's flow law.

*Agree. We corrected the revised manuscript.*

L34: This sounds like radar measurements are the only field method that exists to assess water content. Either pronounce that radar is an example or list other methods. (I'd prefer the former.)

*We agree. We now list other methods and their limitations to assess water content. (lines 35-41)*

L47: If you make 'borehole inclinometry' the subject of the sentence, it becomes much easier to read.

*We modified the revised manuscript accordingly.*

L56: 'Since stresses cannot be measured, …' That is formulated too general. Do you mean for the borehole case?

*Yes, we clarified this in the revised manuscript.*

2 Field site and instrumentation

L101-103: I don't understand this. Why do you need to wait 1 month? Do you define your reference profiles to be the ones 1 month after drilling? Is that because the sensors freeze in? Or is that when you started recording? Please make this sentence clearer.

*Yes, initially we defined the reference profiles in Figure 1b to be the ones 1 month after drilling because the measurement is noisy immediately after installation. In fact, the reference profile would be better defined by the average tilt over this first month (between 15/09/2019 and 15/10/2019) to reduce the noise level. Also, during this month, the tension in the cable tends to better align the sensor along the borehole, making the tilt measurement more representative of the borehole shape. The revised manuscript has been clarified and the figure updated using average tilt between 15/09/2019 and 15/10/2019.*

L149: I personally prefer Global Navigation Satellite System (GNSS), because you probably also used Galileo, Glonass and maybe Beidou satellites.

*Ok, we now use GNSS instead of GPS.*

L150: 'The GNSS receiver network ….'

*Done.*

L154-168: These lines are describing data analysis methods and are in my view neither 'field site', nor 'instrumentation' related.

*In the revised manuscript we moved this part in the methods section "GNSS processing". (Lines 138-142)*

3 Methods

L174-184: Contains results and interpretations. But I see that you try to explain why you applied the methods as you did.

*Yes, we wanted to justify our assumptions here and not come back to it later in the result section that focuses on the quantities derived from the inclinometer (du/dz) that is the core of the study.*

L195: Why? Data gaps?

*Yes, we clarified the revised manuscript.*

L214/215: Formatting of exponents. From what do you compute the discharge? Isn't that measured at Argentiere?

*We corrected the exponents. Yes, nothing is computed, it is measured at Argentiere, we deleted this sentence from the revised manuscript.*

L217: Unclear what you mean. Please clarify or leave this sentence out.

*We clarify this sentence in the revised manuscript :*

*"In addition to deformation measurements, the study of ice rheology requires knowledge of the stress field within the ice, which is evaluated in this study using numerical modelling."*

4 Results

L256-258: Nice! I've seen a similar behavior in boreholes at Rhonegletscher, also at an overdeepening.

L262-264: Please quantify.

*Done.*

L265: '… are very small'. How small? If you use quantifying language, make it concrete and state a number. Eg. 5% of the quantity you're interested in.

*Ok, we now quantify those values in the revised manuscript.*

L267-272: Is the shape factor f=0.646 a fit of the SIA to your full Stokes model, or does it come from the parabolic valley ratio?

*It comes from the parabolic valley ratio according to Nye (1965). The revised manuscript has been clarified.*

L276: I had to read this a couple of times. You can make it clearer by saying: '…and find that simulation with constant creep factor A for a given Glen's law exponent n=3,4,5 yield deformation …'

*Yes, we clarified the revised manuscript.*

5 Discussion

L324/325: I can't follow here. You measure that zigzag shape. That is your data, and the data doesn't care if you neglect du/dx. Please clarify.

*Yes, our wording here was misleading. The data is actually the rate of tilt change, not du/dz. What we are saying here is that the zigzag shape in the reconstructed du/dz may not be real, but may come from how the tilt change is converted to du/dz. For instance, the rate of tilt change in the lower part may be due to du/dx rather than du/dz. This has been clarified in the revised manuscript.*

L347/348: Language problem.

*The revised manuscript has been corrected.*

L364: Either …, or …

*Done.*

L365/357: How do you calculate the 0.17% excess water content? Why

*We have estimated the strain heating change that occurs in response to a 30% increase in deformation rate. This corresponds to an increase of about $3 \times 10^{-4}$ W m$^{-3}$, which can produce $2 \times 10^{-3}$ kg m$^{-3}$ of water in one month (time within which the deformation change occurs). This corresponds to an excess of 0.17% of the water content (kg kg$^{-1}$). We have clarified the revised manuscript.*

L387-390: Yes, but wouldn't the increase in meltwater input also increase the interstitial water content due to increasing subglacial water pressures, resulting in enhanced deformation?

*Actually, subglacial water pressure is higher in winter than in summer according to the borehole water pressure measurement at the drilling site. So, it is unlikely that water content is higher in summer due to an increase in meltwater. We added this argument in the discussion (lines 339-340).*

Figures:

Fig.1  a) The blue dots are the most important, but the green dots are plotted on top of them. Make the blue dots larger to improve their visibility.

*Done.*

Fig.4  b) How does the SIA including the shape factor look like?

*We added this curve in figure 4b.*

Fig.6  c) You plot the velocity only to ~40m depth. I assume that is because you integrate du/dz over depth. However, you know the position of your cable reaching the surface and you have a tiltmeter at 20m depth. From this you get the shape of the borehole. Taking the time derivative also gives the velocity as a function of depth u(z). Is there a reason that you don't show this data?

*The reason is that the inclinometer at 20m-depth sometimes shows unrealistic values, so we removed it here. In the revised Figure 6 we now used an average value of du/dz for the inclinometer at 20m depth that we assume constant through time. The deformation at this depth is almost neglectable and this is not influencing the shape of the velocity profiles presented in this figure. Figure 6 now shows the velocity up the surface in the revised manuscript.*

Fig.7  c) Typo in legend? 'Model channel'

*We corrected the legend.*

Tables:

Table 2: I think this table can go into the supplementary information, because it is not necessary to understand the paper. However, a histogram of water content values could be useful. That's much easier to understand than a table.

*We replaced this table with a new figure summarizing this table. We believe it is now much easier to read and understand.*

Supplementary Information

Fig. S1-S3: I'm not sure if these plots are supporting the manuscript. But I'm ok with having them in the supplementary information.

*Ok, we thought it is always good to show the row data to better grasp what are the challenges associated with interpreting them and estimate du/dz from them.*

Fig. S9/S10: Delta t labels hard to read.

*This has now been improved.*

S3: I think with machine error you mean a systematic uncertainty. Is that correct?

*Yes, we now use the term systematic uncertainty in the revised manuscript.*

Fig. S16: Very hard to see orange line.

*We have changed the colour to make this curve easier to see.*

---

## Author Comment (AC2)

**Response to reviewers**

Reviewer comments in black *- Answers in blue italic*

**Reviewer 2 (Manuela Köpfli):**

**Summary**

The manuscript titled "Creep enhancement and sliding in a temperate, hard-bedded alpine glacier" by Roldán-Blasco et al. examines the basal sliding and internal ice deformation of an alpine glacier. The authors observe an increasing creep factor with depth, which they attribute primarily to changes in water content. However, they suggest that seasonal changes are more likely due to variations in stress distribution within the glacier, as neither the increase in water content nor the stress change alone fully accounts for their observations.

The study is based on approximately 20 tiltmeter measurements along a borehole over an 8-month period. These data are complemented by glacier flow velocity measurements from the glacier surface and bed. Additionally, the authors employ a modeling approach to better understand stress distribution within the glacier. By connecting the modeling output with the tilt and discharge measurements, they quantify the Glen's flow law creep factor, investigate the water content within the ice relative to depth, and analyze stress distribution within the glacier.

Finally, they link their findings to the state and evolution of the subglacial water system. Overall, this manuscript presents significant observations regarding Glen's flow law creep factor and the seasonal evolution of the subglacial water system. Minor comments for improvement focus mainly on enhancing readability.

Manuela Köpfli

*We thank the reviewer for her careful reading and correction, which improved the quality of the manuscript.*

**General questions**

- Would you assume to see similar inclinometer measurements for all boreholes, if they would have worked properly and the installation/drilling would have been perfect?

*The purpose of drilling several boreholes was to have a more robust interpretation if the same behaviour was observed in all of them. Given their similar position on the centreline, we would expect similar behaviour, but it is hard to know as the measurements are difficult to interpret. However, as shown in Figure 2 of the manuscript, the average behaviour appears to be similar.*

- I understand that you mainly focus on the tilt measurements. If possible, I would appreciate seeing a figure of the initial and final borehole shape as Figure 1.b but also showing the final borehole shape.

*We have now added the final shape of the BH2 borehole to Figure 1b.*

- In Figure 2, BH3 shows this larger value at depth around -160m. Do you think this is real? If so, do you have any explanation for that since your boreholes BH2 and BH3 are very close?

*This large value is clearly a measurement artifact due to a poorly stabilized tilt measurement. This can be seen in the raw data in Fig. S2 (BH3#6), where a sudden increase in tilt occurred in August 2020, affecting the calculated mean value of du/dz. We now mention this in the revised manuscript (lines 218-219).*

- In Figure 5 you show your data with a daily resolution. Would it be possible to to "zoom in" to the first half of May and have a higher time resolution for velocities and discharge. It would be very interesting to see, if you see a delay between glacier speed up and discharge.

*The speed up events and their relation to discharge have been closely investigated in a recent paper from our group :*

*Togaibekov, A., Gimbert, F., Gilbert, A., and Walpersdorf, A.: Observing and Modeling Short-Term Changes in Basal Friction During Rain-Induced Speed-Ups on an Alpine Glacier, Geophysical Research Letters, 51, e2023GL107999, https://doi.org/10.1029/2023GL107999, 2024.*

*We now refer to this paper in the revised manuscript for more details on the speed up events.*

**Comments Text**

Note: I recognized that you used different ways to write a date: 15th of February, 15th February, 15th February. Maybe you choose one format and keep it through the whole manuscript. I just wrote "date" and the corresponding line.

Title: I would use "bedded" instead of "beded"

*Done.*

L.32-35: I would break the sentence into two sentences.

*Done. This sentence has also been changed following the comments of reviewer 1.*

L.94: date

*Done.*

L.139: date

*Done.*

L.201: To me, the 2020 is a bit misleading. I would exchange it with something like "over one year (2020)" or "over eight months (2020)" or even remove the 2020 completely.

*Done.*

L.205: To me, there are no residuals of the reconstructed velocities. As I understand, the residuals are more between the reconstructed and observed velocities

*Ok, we clarified the sentence in the revised manuscript.*

L.214: I don't really understand the word "threshold" in this context.

*Yes, the sentence was unclear and has been removed as not relevant here.*

L.214-215: unit goes over the line and exponent is not really an exponent (for both units: m a-1 and m3 s-1)

*This sentence has been removed.*

L.223-237: I'm a bit confused by n and m. If they are not the same, could you add a sentence and describe the difference?

*Ok, we added a sentence in the revised manuscript.*

L.228: I would write m italic that it is immediately clear that you are not talking about meters (like L. 235)

*Done.*

L.236: Normally you used Eq. instead of Eqn.

*This has been corrected.*

L.242: I prefer the italic writing of parameters: A(z)

*This has been modified.*

L.246: Maybe change this sentence to: "Averaged deformation rate profiles, computed from tilt measurements (see methods in Section 3.1) at BH2, BH3, and BH4 between February 15 and October 15, 2020, are shown in Figure 2."

*Done.*

L.247: date

*Done.*

L.275: October 1st to 31st (second superscript is missing), date

*Done.*

L.298: Normally, you wrote section starting with a capital letter

*Done.*

L.307: date

*Done.*

L.313: Normally, you wrote section starting with a capital letter and italic du/dz

*Done.*

L.314: Italic du/dz

*Done.*

L.345: For these reasons we do not include Lliboutry and Duval (1985) is not included in Table 2…

*This has been corrected in the revised manuscript.*

L.364: … either a stress … OR a change…

*This has been corrected in the revised manuscript.*

L.397: use % instead of writing percent

*Done.*

**Comments Figures**

Figure 1: Make sure that all blue dots are visible (especially BH1). Maybe use a different color for the label of the GPS station "ARG1" so that it is immediately clear that this is something different and there is no blue dot behind the green dot. Add to the last sentence in the figure caption, why you start your analysis on 15th of February. date (in the figure caption)

*The figure 1 has been modified accordingly and the legend completed.*

Figure 2: Start figure caption with a capital letter. I would rewrite the figure caption as "Average measured deformation rate profiles with monthly minima and maxima at BH2, BH3 and BH4 including the monthly minima and maxima for BH2. The continuous lines show the average measured deformation rate profile at each borehole for the period between the 15th February and the 15th October 2020, and the shadowed region the range between monthly averaged minima and maxima deformation rate values (shown only for BH2). Every symbol represents a tiltmeter." date (in the figure caption)

*The caption has been corrected.*

Figure 4: Labels a), b), c) are missing in the figure. Is in subplot c) W only inferred for the orange markers? Is the assumption for the blue line, C & P (2010), that W is constant. If so, I would change the label of the orange line/markers also to A(z) and only mention the water content W in the figure caption (as you do).

*Ok, the figure 4 has been modified accordingly.*

Figure 5: I would use a different color for the discharge vs basal velocity, just for clarity. Maybe you want to mention again, that you convolved/averaged your data on a daily basis and you did the same for the discharge? Because, I would assume some daily variation in the discharge over the melting season.

*The color of basal velocity has been changed and the fact the data are at daily time scale is mentioned in the caption.*

Figure 6: date (in the figure caption)

*Done.*

Figure 7: It is correct what you are saying in your figure caption but when I first read that, I did not believe that you wanted to say that the friction increases during melting.

To avoid that, I would add that this is due to "efficient drainage" or an "evolved drainage system".

*Ok, we clarified the caption.*

**Comments Supplement**

Figure S9: I would use Δt instead of "Delatat" and du/dz instead of dud and decide if Error should have a capital letter or not (title vs. color bar label)

*The figure has been modified accordingly.*

Figure S10: Again, i would prefer Δt instead of "Delatat" and ud instead of "du"

*The figure has been modified accordingly.*

Figure S11: I'm not sure if you need that many positions after decimal point for the std.

*True, we kept one decimal only.*

Figure S16: The orange line is hard to see. Also the label for (a) is missing. I would suggest making two figures out of the one figure, just take your figure S16a as one figure and add S16b as S17. This allows you to make S16a bigger and easier to understand. Please check the linewidth of the orange line or change the order you are plotting the lines so that blue and orange are in the front.

*Ok, the figures have been modified accordingly.*